# Mitochondria in Alzheimer’s Disease Pathogenesis

**DOI:** 10.3390/life14020196

**Published:** 2024-01-30

**Authors:** Allison B. Reiss, Shelly Gulkarov, Benna Jacob, Ankita Srivastava, Aaron Pinkhasov, Irving H. Gomolin, Mark M. Stecker, Thomas Wisniewski, Joshua De Leon

**Affiliations:** 1Department of Medicine and Biomedical Research Institute, NYU Grossman Long Island School of Medicine, Mineola, NY 11501, USA; shellygulk1234@gmail.com (S.G.); benna.jacob@nyulangone.org (B.J.); ankita.srivastava@nyulangone.org (A.S.); aron.pinkhasov@nyulangone.org (A.P.); irving.gomolin@nyulangone.org (I.H.G.); joshua.deleon@nyulangone.org (J.D.L.); 2The Fresno Institute of Neuroscience, Fresno, CA 93730, USA; mmstecker@gmail.com; 3Center for Cognitive Neurology, Departments of Neurology, Pathology and Psychiatry, New York University Grossman School of Medicine, New York, NY 10016, USA; thomas.wisniewski@nyulangone.org

**Keywords:** Alzheimer’s disease, mitochondria, inflammation, electron transport chain

## Abstract

Alzheimer’s disease (AD) is a progressive and incurable neurodegenerative disorder that primarily affects persons aged 65 years and above. It causes dementia with memory loss and deterioration in thinking and language skills. AD is characterized by specific pathology resulting from the accumulation in the brain of extracellular plaques of amyloid-β and intracellular tangles of phosphorylated tau. The importance of mitochondrial dysfunction in AD pathogenesis, while previously underrecognized, is now more and more appreciated. Mitochondria are an essential organelle involved in cellular bioenergetics and signaling pathways. Mitochondrial processes crucial for synaptic activity such as mitophagy, mitochondrial trafficking, mitochondrial fission, and mitochondrial fusion are dysregulated in the AD brain. Excess fission and fragmentation yield mitochondria with low energy production. Reduced glucose metabolism is also observed in the AD brain with a hypometabolic state, particularly in the temporo-parietal brain regions. This review addresses the multiple ways in which abnormal mitochondrial structure and function contribute to AD. Disruption of the electron transport chain and ATP production are particularly neurotoxic because brain cells have disproportionately high energy demands. In addition, oxidative stress, which is extremely damaging to nerve cells, rises dramatically with mitochondrial dyshomeostasis. Restoring mitochondrial health may be a viable approach to AD treatment.

## 1. Introduction

Alzheimer’s disease (AD) manifests as progressive cognitive decline eventually ending in death. The disease-defining pathological features observed in the brain are the accumulation of extracellular amyloid-β (Aβ) plaques and intracellular neurofibrillary tangles (NFTs) of hyperphosphorylated tau protein [1]. The mainstay FDA-approved drugs for AD treatment offer some symptomatic relief while newer immunotherapies directed against Aβ may slow the rate of cognitive decline modestly [2,3]. There is no cure and since approaches targeting Aβ and tau have shown that these misfolded proteins are likely not causative, attention has shifted to other mechanisms, including those involving mitochondria [4]. Mitochondria are being explored because abnormalities in this organelle are found early in the course of the disease and can lead to many of the neuron-destroying consequences of AD [5,6,7]. The disruption of mitochondrial dynamics leads to mitochondrial fragmentation, generation of reactive oxygen species (ROS) and poor energy production [8]. Defective mitophagy further aggravates this problem, which impedes the ability of the cell to dispose of the damaged mitochondria [9]. This review examines the central role of mitochondria in the healthy neuron and the pathological mechanisms underlying mitochondrial dysfunction in AD. It explores multiple innovative therapeutic strategies with the potential to add to the pipeline of medications addressing the urgent and growing need to slow or halt the inexorable outcome of this disease. Although other reviews have explored mitochondrial function in AD, the rapid rate of change in the field of AD causation and therapeutics combined with recent data on the subtle effects of new anti-amyloid treatments brings a need for a fresh overview of the topic as provided here [10,11].

## 2. Mitochondrial ATP Production and Oxidative Stress in Neurons

### 2.1. Structural Characteristics

Mitochondria are an essential organelle located in the cytoplasm of eukaryotic cells [12]. They are involved in cellular bioenergetic and signaling pathways and metabolic adaptations to keep the cell and organism alive [13]. They are vital for ATP production through oxidative phosphorylation and for maintaining calcium homeostasis [14,15]. Mitochondria are rod-shaped double-membrane structures ranging in length from 0.5 µm to 1 µm [16]. The outer and inner membranes create two compartments: an intermembrane space and an inner membrane space. The inner membrane has numerous folds called cristae that serve to increase surface area and embedded within the cristae are the proteins needed for oxidative phosphorylation and ATP generation. Enclosed in the inner membrane is a mitochondrial matrix that contains the mitochondrial DNA and holds the enzymes of the citric acid cycle and fatty acid degradation.

### 2.2. Energy Production by Mitochondria and Mitochondrial Oxidative Stress

Mitochondria are particularly important in neurons where energy needs are disproportionately high. Neurons use 70–80% of total energy among brain cells, while glial cells use the remainder [17]. Mitochondria supply 93% of ATP at synapses with glycolysis providing only 7% [18,19]. 

In sequential order, five multiprotein complexes (complex I, complex II, coenzyme Q, complex III, cytochrome C, and complex IV) form the electron transport chain (ETC), a chain that creates an electrochemical gradient and releases energy [20]. (Figure 1). The human mitochondrial genome is circularly organized and consists of 13 proteins, 22 transfer RNAs and 2 ribosomal RNAs encoded by 37 genes [21]. Key protein subunits of complexes I–IV of the ETC are encoded by mitochondrial DNA while other subunits are encoded by nuclear DNA [22]. 

The ETC is embedded within the inner membrane of the mitochondria. ATP synthesis through the ETC is driven by the reduced form of nicotinamide adenine dinucleotide (NADH), which is generated from the citric acid cycle and serves as a donor of electrons to complex I. Two electrons from NADH are transferred to ubiquinone [23]. This electron transfer induces the pumping of protons by complex I from the matrix to the intermembrane space, contributing to the membrane potential and energy storage for ATP production. A second entry point for electrons into the ETC is through complex II, where succinate from the citric acid cycle, when oxidized to fumarate, donates 2 electrons to the oxidized form of flavin adenine dinucleotide (FAD) in complex II to generate the reduced form FADH2. Complex II is not a proton pump and does not translocate protons and, consequently, an FADH2 molecule yields less ATP than an NADH molecule. Both complex I and complex II pass electrons to coenzyme Q at the inner mitochondrial membrane and coenzyme Q accepts electrons in pairs, transfers them to complex III, and then to cytochrome c. Once cytochrome c is reduced, it transfers electrons to complex IV (cytochrome c oxidase), where molecular oxygen (O_2_) is reduced to H_2_O. Lastly, complex V is a multi-subunit complex that functions under a rotational motor mechanism to allow for ATP production [24,25].

A result of the process of electron transfer is the formation of reactive oxygen species (ROS), which contributes to oxidative stress in pathological states [26,27,28]. These ROS are produced in the ETC during the oxidative phosphorylation process, in which oxygen is reduced to H_2_O, and during this course, electrons leak and form superoxide which is then converted to hydrogen peroxide [27]. Hydrogen peroxide can release the very destructive hydroxyl radical [25]. The mitochondria generate approximately 90% of cellular ROS and overproduction of ROS can cause damage to DNA, proteins and lipids [27,28,29]. ROS further impairs mitochondria leading to more ROS production as well as increased mitochondrial membrane permeability, and disruption of calcium homeostasis [30]. 

### 2.3. ATP and Oxidative Phosphorylation

There is high demand for ATP in the very metabolically active neurons in the brain and oxidative phosphorylation, occurring in the inner mitochondrial membrane, is the process in which ATP production and ROS generation are linked [31]. Oxidative phosphorylation has been shown to play a role in AD progression, likely through oxidative damage [32,33,34]. Biffi et al. used the Alzheimer’s Disease Neuroimaging Initiative (ADNI) database to gather SNP genotype and baseline MRI results from 740 subjects in four clinical categories (cognitively normal controls, MCI non-converters, MCI converted to AD, AD) [35]. Their analysis found that 105 genes involved in oxidative phosphorylation contributed to clinical manifestations of AD, with a major role for complex I above other complexes. Venkataraman et al. looked at complex I activity using a specific positron emission tomography (PET) probe (18F-2-tert-butyl-4-chloro-5-{6-[2-(2-fluoroethoxy)-ethoxy]-pyridin-3-ylmethoxy}-2H-pyridazin-3-one) in living persons with AD versus healthy controls and found that complex I was lower in those with AD, particularly in the hippocampus, caudate, and thalamus [36]. 

A decline in ATP is noted with oxidative stress in AD neuropathology. Oxidative stress is an early and prominent feature of AD caused by the overproduction and accumulation of ROS, which damages cells. Zhang et al. determined ATP levels in the brains of AD transgenic mice using an ATP bioluminescence assay and found that ATP content in the AD mouse brain was significantly reduced compared to wild-type C57BL/6 mice, suggesting mitochondrial dysfunction [37].

Armand-Ugon et al. quantified the expression of nuclear genes that encode subunits of the mitochondrial complexes in total homogenates from the entorhinal cortex of AD patients and, using qRT-PCR, found decreased expression of ATP5L, ATPD, and ATP50 genes in later stages of AD compared to early stages [38]. Along these lines, Finney et al. performed an artificial intelligence meta-analysis of dysregulated genes in AD subjects compared to non-demented healthy controls using publicly available transcriptomic datasets of the frontal cortex and cerebellum and found that genes involved in mitochondrial energy, ATP, and oxidative phosphorylation pathways were dysregulated in the AD group [39]. Functional network analysis pinpointed two downregulated genes, ATP5L and ATP5H, each of which encodes a subunit of ATP synthase (mitochondrial complex V), as potentially playing a role in AD pathogenesis. Hence, the dysregulation of ATP production in the mitochondria and the decreased expression of its corresponding ATP production genes, such as ATP5L and ATP5H, underscore the relevance of ATP in AD pathogenesis.

### 2.4. Nicotinamide Adenine Dinucleotide (NAD+) and Complex I 

NAD+ is a coenzyme for redox reactions that acts as an electron acceptor. It is a cofactor for glyceraldehyde-3-phosphate dehydrogenase, the enzyme that catalyzes the dehydrogenation of glyceraldehyde-3-phosphate in the glycolysis process. NAD+ is produced from NADH, primarily by mitochondrial ETC complex I. It is important in energy metabolism, for maintaining mitochondrial homeostasis, and in the stress response to oxidative damage [40,41]. When activity of complex I is compromised during mitochondrial damage, NAD+ production is reduced, and excess NADH accumulates. The ratio of NAD+/NADH can be used as a barometer of mitochondrial function [42]. NAD+ depletion is associated with axonal degeneration [43]. NAD+ levels are lower in AD patient brains [44].

Nicotinamide mononucleotide adenylyl transferase 2 (NMNAT2) is the enzyme responsible for the synthesis of NAD in neurons in the brain and is pivotal in maintaining axonal integrity [45]. Levels of NMNAT2 mRNA are lower in the brain in AD patients and mouse AD models, and overexpression of NMNAT2 in cells that produce excess amyloid precursor protein (APP) suppresses amyloid formation by increasing the NAD+/NADH ratio [46]. Sterile alpha and TNR motif-containing protein 1 (SARM1) are multifaceted metabolic sensors that hydrolyze NAD and are sensitive to changes in NAD levels [47]. Upon axonal injury or mitochondrial malfunction, NMNAT2 levels are reduced, and so is the production of NAD+. NMNAT2 loss promotes SARM1 activation, and this combination results in energetic failure in axons [48,49]. SARM1 and NMNAT2 are considered potential AD therapy targets because of their role in the programmed death of axons in neurodegeneration [50]. 

Correction of NAD+ depletion by NAD+ precursor supplementation has been shown to improve cognitive function in animal models of AD [51,52,53]. Hou et al. used an AD mouse model that is DNA repair deficient and emulates features of human AD such as Aβ plaques, tau tangles, synaptic dysfunction, and cognitive impairment [54]. These mice were reported to have a lower NAD+/NADH ratio, and supplementation with nicotinamide riboside (NR) lessened phosphorylated tau protein pathology and oxidative stress and ameliorated neuroinflammation. 

## 3. Mitochondrial Trafficking

Neurons are highly polarized cells that transfer information through a combination of chemical and electrical signals at the synapse, which is distant from the cell body and maintained by axonal transport [55,56]. Mitochondrial trafficking in neurons is a phenomenon in which mitochondria move bidirectionally, with anterograde transport of mitochondria from the cell body to synaptic terminals and retrograde transport of mitochondria from the synaptic terminals to the cell body [57]. Kinesin-1 mediates anterograde transport and cytoplasmic dynein motors monitor retrograde mitochondrial transport [58]. Studies show that retrograde mitochondrial transport is important for removing aged organelles, and disruption of it impacts the function of motor synapses and the homeostatic distribution of mitochondria throughout the neuron [59]. Both anterograde and retrograde mitochondrial movement are important for axonal outgrowth, synaptic plasticity, and neurotransmission [60]. 

## 4. Mitophagy, Mitochondrial Dynamics and AD

### 4.1. Mitophagy 

Mitophagy, which is a mode of autophagy specifically for mitochondria, is the process of selectively degrading damaged or unneeded mitochondria, which is essential for mitochondrial quality control [61,62]. During mitophagy, the extraneous or defective mitochondria are trafficked to the lysosome, where they are degraded by lysosomal enzymes [63]. Various animal and human studies have established the role of impaired mitophagy in AD [64]. Dysfunctional mitochondria lead to the accumulation of excess ROS and the depletion of ATP [65,66]. Impaired mitochondria at distal sites have to be transported to the soma for lysosomal degradation, and this retrograde transport may be impaired in AD [67,68]. Further, poorly functioning lysosomes may be inefficient in clearing misfolded proteins, such as Aβ [69].

### 4.2. Aβ and Tau in Mitophagy and Mitochondrial Movement

The most well-studied causes of mitochondrial dysfunction in AD relate to the toxicity of Aβ and tau. The accumulation of Aβ causes oxidative stress and the production of ROS by mitochondria. The ROS generated then inflicts damage on mitochondria [70,71]. Mitochondrial ROS production promotes tau aggregation [72]. Aβ and tau also interfere with the trafficking of mitochondria to and from the synapse while also fostering mitochondrial fission, leading to synaptic dysfunction [73,74]. 

Aβ is not produced locally in the mitochondria, so mitochondrial Aβ uptake poses an interesting area of study. Petersen et al. found that in rat mitochondria, Aβ is transported via the translocase of the outer membrane machinery [75]. Immunoelectron microscopy after import showed localization of Aβ to mitochondrial cristae. This was similarly found in human cortical brain biopsies, suggesting that this import machinery can be a unique mechanism for Aβ entry into mitochondria. 

Dou and Tan transfected SHSY-5Y human neuroblastoma cells with plasmids harboring mitochondrial outer membrane protein translocase (TOMM)22 and TOMM40 to directly augment mitochondrial Aβ content. They found that increased Aβ content in the mitochondria enhanced mitophagy, and this could be reversed by transfection with a plasmid harboring presequence protease, responsible for Aβ degradation [76]. In a separate study, presequence protease activity was found by Alikhani and colleagues to be significantly lower in mitochondria isolated from brain tissue specimens obtained post-mortem from the temporal region of AD-affected subjects compared to age-matched controls [77].

The tau protein, a microtubule-associated protein important for synaptic plasticity, acts as a promoter of microtubule assembly, a microtubule stabilizer, and an autophagy regulator [78,79]. The tau protein is primarily expressed in neurons, where it is involved in axonal transport [80,81]. Accumulation of misfolded insoluble phosphorylated tau protein in the neuron leads to aggregation into neurofibrillary tangles that are neurotoxic [82,83,84]. Hyperphosphorylation of tau protein reduces its binding affinity to microtubules, causing microtubules to dissemble and interfering with axonal transport of mitochondria and synaptic vesicles [85,86]. Insufficient mitochondrial presence along the axon starves the synapse of ATP and energy and impedes the autophagic clearance of mitochondria in neurons [87,88]. 

Hu et al. examined the association between intracellular tau accumulation, a hallmark of sporadic AD, and mitophagy using the mitochondrial marker proteins cytochrome c oxidase (COX) IV and TOMM20 [89]. Comparing brain homogenates from AD subjects and age-matched controls, Western blotting showed higher levels of COX IV, TOMM20, total tau and phosphorylated tau in the AD patients. Interestingly, only AD subjects with high tau levels had elevations in COX IV and TOMM20, while AD subjects with normal total tau levels had COX IV and TOMM20 expression comparable to non-AD controls. Since high levels of COX IV and TOMM20 may be considered indicators of mitophagy deficits, these results suggest an association between intracellular tau accumulation and mitophagy deficits [90]. 

Cummins et al. studied the effect of tau accumulation on Parkin-dependent mitophagy in both murine neuroblastoma cells and in the nervous system of C. elegans [91]. When the mitochondrial membrane potential dissipates, mitophagy is normally initiated via the serine/threonine kinase PTEN-induced putative kinase 1 (PINK1) and the E3 ubiquitin ligase Parkin in order to eliminate defective mitochondria. In this PINK1/Parkin pathway, activation occurs when cytosolic Parkin translocates to the surface of the mitochondria, dimerizes, and is trans-autophosphorylated. Activated PINK1 then phosphorylates ubiquitin on the outer mitochondrial membrane proteins, leading to the recruitment and partial activation of Parkin. Following binding to phospho-ubiquitin, Parkin can be fully activated by PINK1. A feed-forward loop leads to the ubiquitylation and addition of poly-ubiquitin chains on the surface of damaged mitochondria, targeting them for autophagy and lysosomal degradation. The study used the mito-QC mitophagy reporter to show that tau specifically impaired Parkin recruitment to defective mitochondria by sequestering it in the cytosol in both the cell and nematode models [91]. This work identified a pathological process in which AD conditions of excess tau can prevent the elimination of dysfunctional mitochondria by obstructing the PINK1/Parkin pathway. 

Fang et al. studied the impact of Aβ and tau on mitophagy in a C. elegans model of AD combined with murine models and cell culture experiments and confirmed that Aβ reduces mitophagy while stimulation of mitophagy decreases Aβ [92]. They also showed that transgenic nematodes overexpressing human tau exhibit reduced basal and stress-induced mitophagy. 

### 4.3. Mitochondrial Fission and Fusion

Mitochondrial fission and fusion are both controlled by large guanosine triphosphatases (GTPases) in the dynamin family. The balance between fusion and fission is critical for meeting energy demand, as excess fission leads to mitochondrial fragmentation while excess fusion leads to elongated mitochondria with high levels of ROS [93]. The specific proteins that regulate fission include dynamin-1-related protein (DRP)1, fission (Fis)1, and mitochondrial fission factor (Mff), while mitochondrial fusion is regulated by proteins such as mitofusin 1 (Mfn1), mitofusin 2 (Mfn2), and optic atrophy protein (OPA1) [94]. 

While mitochondrial fission is required for both mitophagy and mitochondrial transport, excess fission leads to fragmentation, and patients with AD are determined to have higher expression of mitochondrial fission genes such as DRP1 [95,96,97,98]. Enhanced fission leads to structural damage to mitochondria in neurons in the AD brain [99,100]. Upregulation of fusion proteins or interference with fission proteins may rescue neurons from the consequences of overzealous fission [101,102,103]. 

Using rat primary hippocampal neurons, Li et al. showed that abnormal mitochondrial fusion is also destructive [104]. Overexpression of tau protein in the cultured rat neurons increased fusion proteins, leading to mitochondrial elongation and decreased viability of neurons. 

### 4.4. Effects of Amyloid and Tau on Fission and Fusion 

Drp1 may contribute to the pathogenesis of AD by interacting with Aβ and phosphorylated tau, leading to excessive mitochondrial fragmentation with negative consequences such as synaptic dysfunction and neuronal damage [105]. 

Glycogen synthase kinase 3 β (GSK3β), primarily present in the brain and activated by Aβ, is responsible for phosphorylation of tau and is considered a crucial enzyme in the pathobiology of AD [106,107]. In murine models, overexpression of GSK3β increases tau phosphorylation and promotes disassembly of microtubules [108]. GSK3β also phosphorylates Drp1 at multiple serines, affecting mitochondrial fission and fragmentation [109,110]. Yan et al. showed that inhibition of GSK3β can impede mitochondrial fragmentation and confer neuroprotection in both transgenic amyloid precursor protein/presenilin 1 (APP/PS1) AD mice and cultured rat primary hippocampal neurons via prevention of Drp1 phosphorylation [111].

GSK3β also enhances the formation of Aβ by inducing β-site APP-cleaving enzyme (BACE)1, a critical enzyme in the amyloidogenic pathway that converts amyloid precursor protein (APP) to Aβ [112]. GSK3β effects on BACE1 are dependent on NF-κB signaling.

PINK1, key in mitophagy initiation, can help to maintain mitochondrial integrity, control oxidative stress, enhance Aβ clearance, and improve learning and memory in AD rodent models and human neuronal cybrids carrying AD-derived mitochondria [92,113,114]. PINK1 has the ability to phosphorylate Drp1, thereby affecting mitochondrial dynamics by promoting fission, while Parkin can ubiquitinate Drp1 to promote its proteasomal degradation [115,116]. The ability of PINK1 to affect mitochondrial fragmentation in neurons via Drp1 leaves open an avenue for possible AD therapy by regulating the expression of PINK1 as a means of controlling fission and fragmentation. 

## 5. Mitochondrial DNA Methylation

Mitochondrial DNA methylation, a mechanism of epigenetic control, is a suspected contributing factor in AD pathogenesis [117,118]. Xu and colleagues compared mitochondrial DNA methylation in the hippocampi of transgenic APP/PS1 AD mice to age-matched wild-type C57BL/6J mice and found hypomethylation of the D-loop region (critical for mitochondrial DNA replication and transcription) and hypermethylation of the 12 S rRNA gene in the hippocampi of the AD mouse model [119]. The AD mice also showed a decrease in mitochondrial DNA copy number and lower gene expression compared to the C57BL/6J mice, indicative of mitochondrial dysfunction and abnormal biogenesis [120,121]. In a later study, Xu et al. examined methylation in mitochondrial cytochrome b (CYTB) and COX II in the hippocampi of APP/PS1 AD mice in comparison to C57BL/6J mice and found hypermethylation and decreased mitochondrial DNA copy numbers in the hippocampus of these specific genes in the AD transgenic mice [122]. Impaired COX function has been related to ROS production [123,124]. 

Ding et al. isolated cell-free DNA from blood samples taken from 31 AD patients and 26 age- and sex-matched controls and compared mitochondrial DNA methylation patterns between the groups [125]. They found excess hypomethylation of mitochondrial DNA in AD patients compared with controls, with hypomethylation primarily located in non-protein-coding regions of mitochondria.

In human brain specimens from persons of advanced age, Klein et al. observed a decrease in quantity of mitochondrial DNA in AD versus non-AD controls when sequencing 1361 samples, and more specifically, found a 7–14% decrease in mitochondrial DNA copy number in AD compared to non-AD controls [126].

## 6. Glucose Metabolism Reduced in AD

The human brain is one of the most metabolically active organs in the body and predominantly utilizes glucose as its main energy source. The brain is relatively inflexible in using alternative substrates aside from glucose for energy production [127,128]. AD is characterized by reduced glucose metabolism in the brain [129,130]. This hypometabolism is detectable in PET scans and serves as an early imaging modality for AD detection and prediction of progression from MCI to AD [131,132,133,134]. A longitudinal study of cognitively normal older persons and persons with mild AD showed a progressive reduction in glucose utilization prior to dementia onset in those who began the study without cognitive symptoms [135]. In the early stages of AD, glucose hypometabolism is apparent in the hippocampal and posterior cingulate of the human brain, areas that also show abnormal patterns of functional connectivity early in AD [136,137,138]. As the disease advances, glucose consumption is reduced at the temporal–parietal cortex and the frontal and occipital cortices [139]. Hypometabolism is generally bilateral in AD but may be left lateralized in early MCI [140].

Hypometabolism may be due partly to reduced glucose transport at the blood–brain barrier and across astrocytic and neuronal cell membranes. The transport of glucose from the bloodstream to the parenchymal cells is facilitated by integral membrane proteins called glucose transporters (GLUTs). These sodium-independent facilitative transporters play an important role in glucose metabolism [141]. The majority of glucose uptake in the brain occurs via GLUT1 and GLUT3. While GLUT1 moves glucose across the blood–brain barrier into astrocytes, Glut3 handles the majority of glucose uptake by neurons [142,143]. Decreased levels of GLUT1 and GLUT3 are particularly seen in the cerebral cortex and hippocampus of AD patients, with significant loss of GLUT3 [144]. One proposed mechanism for decreased levels of GLUT1 and GLUT3 in the AD brain is the downregulation of hypoxia-inducible factor-1 (HIF-1) [145]. HIF-1 suppression subsequently causes abnormal tau phosphorylation and/or neurofibrillary degeneration by downregulating the hexosamine biosynthesis pathway. [146]. Another putative mechanism contributing to decreased GLUT3 expression involves the transcription factor cAMP response element (CRE)-binding protein (CREB), known to be important in supporting cognition, memory formation, and neuronal survival [147]. The human GLUT3 promoter has three potential (CRE)-like elements where CREB can bind and induce GLUT3 expression. In AD, CREB level is low in the hippocampus [148]. Jin et al. found decreased expression of full-length CREB and increased CREB truncation in the AD brain. CREB truncation is linked to activation of calpain 1, which proteolyses CREB, thus leading to a deficit in CREB and reduced GLUT3 expression [149]. 

Studies have also explored the effects of reduced O-GlcNAcylation, a posttranslational modification that regulates human nuclear, cytoplasmic, and mitochondrial proteins in AD [150]. O-GlcNAcylation involves the attachment of a single N-acetylglucosamine sugar to specific serine or threonine residues on over 9000 proteins of nuclear, cytosolic, and mitochondrial origin [151]. Evidence from multiple laboratories supports the hypothesis that dysregulated O-GlcNAcylation contributes to AD, but the mechanisms and specifics of which proteins are impacted remain unresolved [152,153]. O-GlcNAcylation of brain proteins, including tau, is significantly decreased due to impaired glucose metabolism in the AD brain [154]. In both cell culture and murine models, Pinho et al. found that enhanced O-GlcNAcylation led to improved mitochondrial network and cell viability, while Park et al. observed that augmenting O-GlcNAcylation in brain tissue in a mouse AD model lowered ROS levels and improved the morphology of mitochondria [155]. Drugs are being tested for their ability to increase O-GlcNAcylation of tau protein to prevent hyperphosphorylation and NFT formation [156,157]. In contrast, prolonged elevations of O-GlcNAcylation may interfere with mitochondrial ATP production [158]. However, how diminished or enhanced O-GlcNAcylation directly affects mitochondrial and neuronal functions in the context of AD still needs to be elucidated [159]. 

## 7. Apolipoprotein (Apo)E Gene Impact on Mitochondria and Bioenergetics

ApoE is a 299 amino acid lipid transport protein and the most abundant brain apolipoprotein. ApoE functions to maintain brain lipid balance and facilitates the exchange of lipids between neurons and glial cells. Its expression in the brain is upregulated in activated microglia and in stressed neurons as an adaptation to inflammatory and cellular stress conditions [160,161]. 

There are three predominant ApoE isoforms in humans: ApoE2, ApoE3, and ApoE4, which are the products of the ε2, ε3, and ε4 alleles, respectively. These confer varying degrees of AD risk, with the ApoE ε4 allele being the strongest genetic risk factor for sporadic AD, while the ApoE ε2 allele is associated with the lowest AD risk [162,163,164]. ApoE4 is expressed in more than half of AD patients, and its prevalence makes it an important therapeutic target [165,166]. In addition to its role in binding and clearing amyloid-β, ApoE also affects mitochondria [167]. The ApoE ε4 allele causes mitochondrial dysfunction and alters mitochondrial-associated membranes, key membranes that connect the endoplasmic reticulum with mitochondria [168,169]. Mitochondrial-associated membrane disruption is found in the context of AD and can negatively influence calcium and lipid metabolism [170,171].

In a study by Simonovitch and colleagues, female mice were generated by replacing endogenous murine ApoE with either human APOE3 or APOE4, and then the mitochondria were characterized [172]. The researchers found that, compared to mice expressing ApoE3, mice expressing APOE4 had elevated hippocampal levels of mitochondrial fusion-mediating protein (MFN)1 combined with reduced levels of dynamin-related protein 1 (Drp1), a critical protein that controls mitochondrial fission. Transmission electron microscopy showed abnormal morphology in the APOE4 mice, with elongated mitochondria and less dense cristae. Levels of the mitochondrial ETC protein COX1 in hippocampal neurons were enhanced in the ApoE4 mice, possibly as a compensatory mechanism.

ApoE-ε4 carriers show decreases in mitochondrial respiratory complexes in neurons, which supports the early role of energy metabolism and the progression of AD [173]. Yin et al. performed a study on postmortem human brain tissue and measured proteins that are responsible for mitochondrial biogenesis in 46 cases, including ApoE-ε4 carriers (n = 21) and non-carriers (n = 25) [174]. These patients had undergone clinical and neuropsychological assessments prior to death that were then correlated to brain tissue protein data. The results showed that harboring an ApoE-ε4 allele was associated with decreased levels of antioxidative stress and synaptic plasticity proteins. 

Costa-Laparra et al. studied the impact of the Apoe ε4 allele co-occurring with a presenilin 1 (PSEN1) gene mutation by analyzing skin fibroblasts from AD patients harboring this combination compared to fibroblasts from persons with 2 ApoE ε 4 alleles and healthy ApoE3/3 controls [175]. They observed that cells that were ApoE3/4 + PSEN1 had lower viability, an accumulation of lysosomes, and greater vulnerability to oxidative stress than either homozygous ApoE4 cells or homozygous control ApoE3 cells. Homozygosity for the APOE ε4 allele alone led to increased mitochondrial fragmentation, while the PSEN1 mutation alone caused impairment of mitochondrial network integrity. Orr et al. used N2a mouse neuroblastoma cells stably expressing ApoE3 or ApoE4 to show mitochondrial dysfunction attributed specifically to the ApoE4 gene, including a lower NAD+/NADH ratio, higher levels of ROS, and reduced ATP generation capacity [176]. This suggests that ApoE4 impairs mitochondrial respiration.

Lee et al. looked at Apoe4-induced lysosomal cholesterol accumulation in astrocytes and the resulting impairment of lysosome-dependent removal of damaged mitochondria [177]. They found that, compared to ApoE3 astrocytes, ApoE4 astrocytes had impaired autophagy, increased ROS, and enhanced glycolysis. Cholesterol-depleting agents restored autophagy and mitochondrial respiration but did not normalize glycolysis.

## 8. Mitochondria in the Treatment of AD

The importance of mitochondrial dysfunction in the pathogenesis of AD has prompted the exploration of new treatment strategies designed to improve mitochondrial function [178,179,180]. A major focus has been on correcting oxidative stress imbalances using mitochondrial-targeted antioxidant therapies, and many studies have been conducted in rodent models showing efficacy [181,182,183]. 

Mitoquinone mesylate (MitoQ, 10-(4,5-dimethoxy-2-methyl-3,6-dioxo-1,4-cyclohexadienlyl) decyl triphenylphosphonium methanesulfonate)) is a compound composed of a derivative of ubiquinone targeted to mitochondria by covalent attachment to a lipophilic triphenylphosphonium, which facilitates crossing of the molecule through the layers of the mitochondrial membranes. The ubiquinone can then be converted to the antioxidant ubiquinol by complex II of the ETC. MitoQ behaves as a scavenger for ROS and has been previously tested in AD nematode and mouse model systems [184,185,186]. In these models, MitoQ has been shown to prevent oxidative damage, reduce Aβ accumulation, astrogliosis, synaptic loss, and improve cognitive function. It is considered a dietary supplement, can be taken orally, and crosses the blood–brain barrier. Human studies have shown a benefit in improving vascular function in older persons [187], and a human trial entitled “The Mito-Frail Trial: Effects of MitoQ on Vasodilation, Mobility and Cognitive Performance in Frail Older Adults (Mito-Frail)” is about to begin (https://classic.clinicaltrials.gov/ct2/show/NCT06027554, accessed on 4 January 2024). Plastoquinonyl decyl triphenylphosphonium (SkQ1), a derivative of plastoquinone, is similar to MitoQ in that it is a mitochondria-targeted antioxidant that shows neuroprotection in murine models [188].

Other antioxidant compounds such as mito-apocynin, made from apocynin, a plant-derived inhibitor of NADPH (nicotinamide adenine dinucleotide phosphate) oxidase, and astaxanthin, a red pigment with potent antioxidant properties, have also shown potential for improving mitochondrial dysfunction in preclinical models and could be used in humans in the future [189,190,191,192,193]. 

Mitochondrial fragmentation is detrimental to cellular bioenergetics. As discussed earlier, the mitochondrial fission protein Drp1 is abnormally expressed in AD, leading to excess mitochondrial fragmentation [194]. Drp1 also interacts with Aβ and hyperphosphorylated tau and promotes changes in mitochondrial morphology and bioenergetics, negatively impacting ATP production. Interactions between Drp1 and Aβ induce synaptic loss [195,196]. The Drp1 inhibitor, mitochondria division inhibitor 1 (Mdivi1), a quinazolinone derivative, has been found to effectively target synaptic depression that occurs due to Aβ in AD [197,198]. Mdivi1 has been found to specifically target mitochondrial dysfunction by attenuating ROS production and enhancing ATP production [199]. These findings suggest that Mdivi1 is a potential therapeutic option for treating mitochondrial dysfunction and synaptic depression associated with Aβ-induced pathology in hippocampal cells in AD [200,201]. However, it has drawbacks, including a lack of specificity and a propensity to aggregate, that make it likely that better compounds can be developed for human use [202].

Diethyl (3,4-dihydroxyphenethylamino) (quinoline-4-yl)methyl phosphonate (DDQ) is a pharmacologically developed compound that can cross the blood–brain barrier and has shown positive effects on mitochondrial dysfunction and synaptic dysregulation at both mRNA and protein levels [203,204]. DDQ reduces the fission proteins Drp1 and Fis1 while increasing the fusion proteins Mfn1 and Mfn2. It reduces the interactions of DRP1 with Aβ, inhibiting Aβ-DRP1 complex formations of [205]. Aβ-Drp1 complexes are known to promote mitochondrial fragmentation, mitochondrial DNA mutations, and a reduction in mitochondrial oxidative phosphorylation, all of which are observed in AD brains. Drp1 inhibition has emerged as a therapeutic target because it has been shown to improve learning and memory and protect mitochondria from fragmentation in mouse models of AD [206,207]. 

Nicotinamide compounds such as nicotinamide mononucleoside, nicotinamide mononucleotide, and nicotinamide riboside are being evaluated for their effect on mitophagy and NAD levels [54,208,209]. Although not curative, this type of dietary supplementation may be part of a multi-faceted approach to AD treatment. 

Overall, these treatments are designed to improve mitochondrial robustness in order to minimize oxidative stress while maintaining ATP production (Table 1). Although new therapies have long development times, targeting therapeutics aimed at mitochondrial dysfunction has shown promising pre-clinical effectiveness. However, a better understanding of the various signaling networks formed by mitochondria within neurons can pave the way for the development of more disease-modifying therapies. 

## 9. Limitations of the Hypothesis That AD Is Driven by Mitochondrial Dysfunction

There are many arguments to support the involvement of mitochondria in clinical human AD. However, because of the protean manifestations of AD, the question arises as to what proportion of the association between mitochondrial failure and AD is secondary to other processes that produce alterations in both mitochondria and cognitive function. In order to have a well-rounded picture, it is also important to understand some of the potential problems associated with the hypothesis that mitochondrial pathology may be the cause of AD. This provides the nidus for ideas to spur future research. 

Traditional mitochondrial disorders caused by mutations in mitochondrial DNA, such as mitochondrial encephalopathy, myopathy, lactic acidosis, and stroke-like episodes (MELAS) and myclonus epilepsy with ragged-red fibers (MERRF), among others, may be associated with cognitive problems but may also confer symptoms such as myopathy, epilepsy, myoclonus, and lactic acidosis [210]. Leber’s hereditary optic neuropathy (LHON) is generally restricted to the eye with progressive bilateral loss of vision [211]. Although epilepsy, myoclonus, and retinal problems may be seen in AD, they are far less frequent or dramatic than with diseases such as MERFF or MELAS [212,213,214]. Cranial neuropathy, myopathy, and lactic acidosis are not common in AD. 

Traditional mitochondrial disorders may be associated with white matter abnormalities and cerebellar atrophy on MRI [215]. Although cerebellar atrophy and white matter abnormalities may be observed in AD as well, they are not as prominent as cortical atrophy [216,217].

Some have claimed that both Aβ accumulation and mitochondrial failure are critical to the development of AD and may act synergistically [218,219]. The problem with this theory is that most of the Aβ deposition is extracellular, although some argue for the role of intraneuronal Aβ [220]. It is possible that Aβ oligomers may be at increased concentration near the mitochondria, but it would have to be demonstrated that this occurs early in the disease. It is also possible that some of the studies showing reduced mitochondrial biochemical output could be spurious and might be secondary to atrophy and loss of neurons rather than primary. The basic conundrum is that mitochondria are critical for respiration throughout the body, while AD is a disease of the cerebrum. If the primary pathologic mechanism of AD is mitochondrial, why are problems in other organs and systems not prominent? 

## 10. Conclusions

Mitochondrial dysfunction plays a critical role in multiple aspects of the development of AD (Figure 2). Failure of mitochondria leads to insufficient energy supply and oxidative stress, which further erodes mitochondrial integrity and damages the neuron, particularly the axon. Restoring and maintaining mitochondrial health is increasingly the focus of investigation as a therapeutic strategy in AD, particularly in light of the inability of anti-amyloid and anti-tau treatments to halt AD progression. An intensive study is required to gain a better understanding of the underlying mechanisms. Indeed, recent research in the field has demonstrated the involvement of impaired mitochondrial dynamics, biogenesis, and mitophagy in AD, which offers multiple sites and pathways to target novel interventions directed at sustaining the integrity of mitochondria. In vitro and in vivo studies on genetic models of AD demonstrate a role of APP or Aβ in impairment of mitochondria, opening up the possibility of multi-targeted treatments aimed at both optimizing bioenergetics and reducing amyloidogenesis. 

## Figures and Tables

**Figure 1 life-14-00196-f001:**
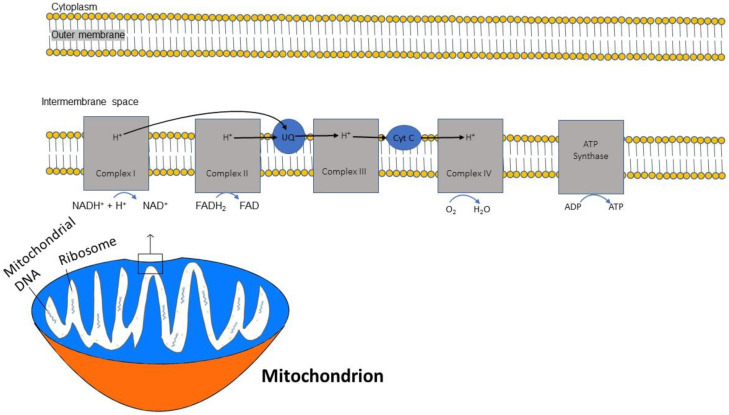
Diagram demonstrating the flow of electrons through the mitochondrial electron transport chain (ETC). Electrons initially enter the ETC by NADH at complex I, and FADH2 from complex II. Ubiquinone transports the electrons to complex III, and then through cytochrome (Cyt) C to complex IV where oxygen is reduced into water. A proton gradient pumped across the inner mitochondrial membrane caused by the translocation of protons synthesizes ATP.

**Figure 2 life-14-00196-f002:**
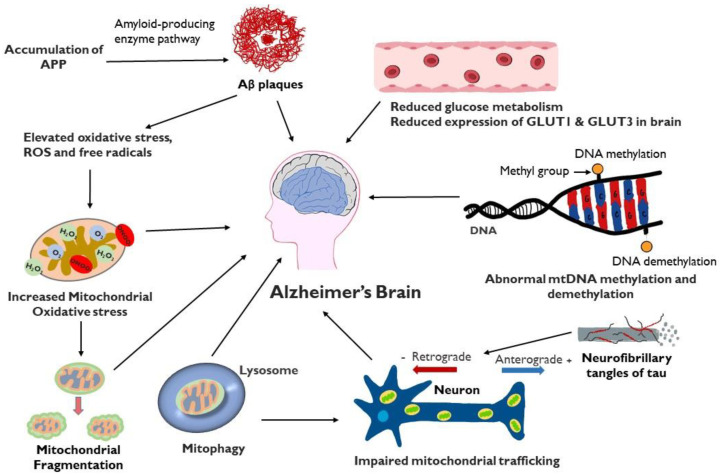
Schematic representation of factors involved in mitochondrial dysfunction and AD: Mitochondrial dysfunction in the AD brain results in elevated oxidative stress, increased mitochondrial fragmentation, mitophagy, impaired mitochondrial trafficking, mitochondrial DNA damage, defective mitochondrial biogenesis and dynamics and reduced glucose metabolism. Accumulation of Aβ can contribute to mitochondrial oxidative stress while tau protein, present in extracellular tangles, can interfere with axonal movement of mitochondria.

**Table 1 life-14-00196-t001:** Therapies for AD targeting mitochondrial function.

Treatment	Effect on mitochondria and neuron	References
MitoQ	Prevention of ↑ ROS production, ↓ Aβ accumulation, ↓ astrogliosis, minimize synaptic loss	[184,185,186]
SkQ1	↓ Aβ accumulation and tau hyperphosphorylation in hippocampus in rat AD model	[188]
Mito-apocynin	NADPH oxidase inhibitor that acts as an anti-inflammatory and antioxidant.	[189,190]
Astaxanthin	Carotenoid and dietary supplement with neuroprotective and antioxidant effects. Maintains mitochondrial membrane potential.	[191,192,193]
Mdivi1	↓ in the fission proteins Drp1 and Fis1, ↑ in the fusion proteins Mfn1 and Mfn2, ↓ excessive fragmentation of mitochondria, inhibition of Aβ-DRP1 complex formation	[197,198,199,200,201]
Nicotinamide compounds: nicotinamide mononucleoside, nicotinamide mononucleotide, nicotinamide riboside	↓ DNA damage, ↓ neuroinflammation, and ↓apoptosis of hippocampal neurons	[209]

Abbreviations: Aβ—amyloid β; DRP1—dynamin-1-related protein 1; Mdivi1—mitochondria division inhibitor 1; Mfn—mitofusin; MitoQ—mitoquinone mesylate; NADPH—nicotinamide adenine dinucleotide phosphate; ROS—reactive oxygen species; SkQ1—plastoquinonyl decyl triphenylphosphonium; ↑—increased; ↓—decreased.

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
