# Peer review of "Mitochondria in Alzheimer’s Disease Pathogenesis"

_life, 2024, doi:10.3390/life14020196_

Round 1
Reviewer 1 Report
Comments and Suggestions for Authors
The review article entitled "Mitochondria in Alzheimer's Disease Pathogenesis" by Reiss et al. covers many of the latest findings and is very well to understant and meaningful. I strongly recommend publicaion in Life.
The main change requested is that Figure 1 is a diagram of the general mitochondrial electron transport chain, which seems to have little relevance to the purpose of this review. This figure 1 is not important. The authors review a wide range of topics, including mitochondrial trafficking, mitophagy, the relationship with Ab and Tau, mtDNA methylation, glucose metabolism, ApoE, and therapeutic drugs. Therefore, I strongly recommended that these topics should be changed to a schematic diagram summarizing the points discussed in the review, such as mitochondria as primary and secondary contributors to Alzheimer's disease. As an example, I think it would be great to include therapeutic drugs in addition to the schematic diagrams summarized in a related review, https://molecularneurodegeneration.biomedcentral.com/articles/10.1186/s13024-020-00376-6.
Author Response
We thank the reviewer for thoroughly scrutinizing our manuscript. As requested, we have revised the manuscript and addressed the specific comments of the reviewer. The revised sections are delineated in red in a marked copy of the manuscript text.
Below, we provide a point-by-point response to each of the reviewer’s comments.
Reviewer # 1 Comments
- COMMENT #1: The main change requested is that Figure 1 is a diagram of the general mitochondrial electron transport chain, which seems to have little relevance to the purpose of this review. This figure 1 is not important. The authors review a wide range of topics, including mitochondrial trafficking, mitophagy, the relationship with Ab and Tau, mtDNA methylation, glucose metabolism, ApoE, and therapeutic drugs. Therefore, I strongly recommended that these topics should be changed to a schematic diagram summarizing the points discussed in the review, such as mitochondria as primary and secondary contributors to Alzheimer's disease. As an example, I think it would be great to include therapeutic drugs in addition to the schematic diagrams summarized in a related review, https://molecularneurodegeneration.biomedcentral.com/articles/10.1186/s13024-020-00376-6.
RESPONSE: We have added a new figure that presents a schematic diagram of the points discussed. We have retained the electron transport chain figure as we believe it is a useful reference for our discussion of the complexes and makes the narrative easier to follow.
We thank the reviewer and believe that the manuscript is improved as a result of their input. We hope you will decide in favor of accepting our report at this time.
Sincerely,
Allison B. Reiss, M.D.

Reviewer 2 Report
Comments and Suggestions for Authors
The review article: "Mitochondria in Alzheimer's disease pathogenesis" by Riess et al. is an interesting and very important review of Alzheimer's disease pathogenesis.
General concept comments
-Overall, this is a paper properly written with clear and professional language. However, please double-check the grammar and the terminology.
-This review summarizes the current literature on "Mitochondria in Alzheimer's disease pathogenesis" and provides current references, but some papers from as early as 1994 (Ref 141) and 1997 (Ref 164) are referenced, which were published about 30 years ago. Please use more recent references for those papers.
Specific comments
-This is an important review, please justify clearly in the manuscript how this review is different and more important than those already in the literature, what new is being offered by this work, and what is different from the other review, such as Wang et al. (2020).
Validity of the findings
-There is enough evidence provided to support the findings. The conclusion also summarizes the findings and suggests specific future research and treatment directions.
Comments on the Quality of English LanguageThis paper is written properly, clearly, and in professional language. However, please double-check all grammar and terminology.
Author Response
We thank the reviewer for thoroughly scrutinizing our manuscript. As requested, we have revised the manuscript and addressed the specific comments of the reviewer. The revised sections are delineated in red in a marked copy of the manuscript text.
Below, we provide a point-by-point response to each of the reviewer’s comments.
Reviewer # 2 Comments
- COMMENT #1: This review summarizes the current literature on "Mitochondria in Alzheimer's disease pathogenesis" and provides current references, but some papers from as early as 1994 (Ref 141) and 1997 (Ref 164) are referenced, which were published about 30 years ago. Please use more recent references for those papers.
RESPONSE: We have replaced the references with more recent ones. Due to other edits, reference 141 is now reference 144 and reference 164 is now reference 167.
- COMMENT #2: This is an important review, please justify clearly in the manuscript how this review is different and more important than those already in the literature, what new is being offered by this work, and what is different from the other review, such as Wang et al. (2020).
RESPONSE: We have added the justification which stems from the rapidity of change in the field, especially with the approval of anti-amyloid therapies that have not had the groundbreaking effect that was hoped for.
We have added the following justification to the introduction along with 2 new references: “Although other reviews have explored mitochondrial function in AD, the rapid rate of change in the field of AD causation and therapeutics combined with recent data on subtle effects of new anti-amyloid treatments brings a need for a fresh overview of the topic as provided here.”
- COMMENT #3: There is enough evidence provided to support the findings. The conclusion also summarizes the findings and suggests specific future research and treatment directions.
RESPONSE: We are very glad that the reviewer finds our review to be well-supported and forward-looking toward a future of better therapies for AD.
We thank the reviewer and believe that the manuscript is improved as a result of their input. We hope you will decide in favor of accepting our report at this time.
Sincerely,
Allison B. Reiss, M.D.
